# Standardized Method for the Assessment of Behavioral Responses of Zebrafish Larvae

**DOI:** 10.3390/biomedicines9080884

**Published:** 2021-07-24

**Authors:** Hideyuki Maeda, Noritoshi Fukushima, Akihiro Hasumi

**Affiliations:** 1Department of Forensic Medicine, Tokyo Medical University, 6-1-1 Shinjuku, Shinjuku-ku, Tokyo 160-8402, Japan; akihirohasumi@gmail.com; 2Department of Preventive Medicine and Public Health, Tokyo Medical University, 6-1-1 Shinjuku, Shinjuku-ku, Tokyo 160-8402, Japan; fukufuku@tokyo-med.ac.jp

**Keywords:** abnormal behavior, arrhythmia, behavioral analysis, high-throughput screening, light–dark test, toxicology, zebrafish larvae

## Abstract

Zebrafish are easy to breed in a laboratory setting as they are extremely fertile and produce dozens of eggs per set. Because zebrafish eggs and the skin of the early-stage larvae are transparent, their embryos and the hearts and muscles of their larvae can be easily observed. Multiple rapid analyses of heart rate and behavior can be performed on these larvae simultaneously, enabling investigation of the influence of neuroactive substances on abnormal behavior, death, and associated pathogenetic mechanisms. Zebrafish larvae are becoming increasingly popular among researchers and are used in laboratories worldwide to study various vertebrate life phenomena; more experimental systems using zebrafish will undoubtedly be developed in the future. However, based on the available literature, we believe that the conceptualization of a protocol based on scientific evidence is necessary to achieve standardization. We exposed zebrafish larvae at 6–7 days post-fertilization to 50 repeated light–dark stimuli at either 15-min or 5-min intervals. We measured the traveled distance and habituation time through a video tracking apparatus. The traveled distance stabilized after the 16th repetition when the zebrafish were exposed to light–dark stimuli at 15-min intervals and after the 5th repetition when exposed at 5-min intervals. Additionally, at 15-min intervals, the peak of the traveled distance was reached within the first minute in a dark environment, whereas at 5-min intervals, it did not reach the peak even after 5 min. The traveled distance was more stable at 5-min intervals of light/dark stimuli than at 15-min intervals. Therefore, if one acclimatizes zebrafish larvae for 1 h and collects data from the 5th repetition of light/dark stimuli at intervals of 5 min in the light/dark test, a stable traveled distance result can be obtained. The establishment of this standardized method would be beneficial for investigating substances of unknown lethal concentration.

## 1. Introduction

Zebrafish (*Dario rerio*) have various practical advantages that make them useful animal models, such as small size, optical transparency, and high fertility, which makes them easy to breed and allows for in vitro fertilization [1]. The morphology and biology of zebrafish are also sometimes homologous to those of mammalian species, making them an attractive animal model for studying human disorders [2]. The zebrafish model is further validated by their heredity and behavior. Zebrafish exhibit a variety of complex behaviors, including anxiety, startling, and defensive behaviors that can help model learning and facilitate the study of memory formation and neurological and psychiatric disorders [1]. For instance, a natural preference for dark environments (scototaxis) in adult zebrafish is a useful indicator of anxiety, which is reduced by anxiolytic drugs and increased by anxiogenic agents [3]. Because of their short generation time, permeability to small molecules, similarity of morphology, genes, and behaviors to other vertebrates, and ease of manipulation, the locomotor activity of zebrafish larvae in response to certain stimuli can be studied through high-throughput automated video-tracking analyses [4]. The light–dark locomotion test has been proposed as a useful screening method for psychoactive substances. In this test, zebrafish larvae are placed in multi-well plates inside a closed chamber and exposed to alternating light and dark conditions; then, their locomotor activity is measured through a high-throughput video-tracking analysis [1]. Light–dark transitions reduce the locomotor activity of zebrafish larvae, whereas light–dark transitions have the opposite effect, reflecting increased stress/anxiety levels [5].

While adult zebrafish require 4.5 L of water per animal [6], responses of zebrafish larvae to light and dark stimuli can be observed with only 0.3 mL of water per animal [1]. Moreover, it is possible to study the effects of smaller amounts of drugs on behavior using zebrafish larvae [7]. Therefore, zebrafish larvae can greatly contribute to the pathophysiology analysis and large-scale screening of drugs and toxic substances that can be lethal or cause organ injury and abnormal behaviors. Basnet et al. (2019) summarized experiments in which drugs were used to stimulate responses to light and dark conditions in zebrafish larvae, and revealed that the experimental conditions and target drugs/toxicants varied, highlighting the need for standardization [1]. Although several papers have analyzed behavioral responses to light stimuli, none have thoroughly examined the effect of habituation time to the experimental conditions, intervals between light on/off, and the number of stimuli. The aim of this study was to determine an experimental method capable of analyzing the responses of zebrafish larvae to light and dark stimuli.

## 2. Materials and Methods

### 2.1. Study Animals and Husbandry

Adult zebrafish (Danio rerio; undefined wild-line, obtained from Aquatic Research Organisms, Inc., Hampton, NH, USA) were reared and maintained at a density of 3 fish per liter in a small automatic aquarium (approximately 7 L) under the standard conditions of a recirculated water system supplied with dechlorinated urban tap water. This water had been subjected to reverse osmosis and was maintained at pH 7.5, a water temperature of 27 ± 1.0 °C, and electrical conductivity of 400 ± 50 S/m. Lighting was artificial with a sequence of 14 h of light (8 a.m. to 10 p.m.) and 10 h of darkness.

Two adult male and female zebrafish were placed in a breeding tank in an incubator (IC101, Yamato Scientific Co., Ltd., Tokyo, Japan) set at 28 °C in which a partition was placed during the afternoon and night. The fish were left undisturbed overnight, and the partition was removed the next morning. As soon as the embryos appeared, the adults were returned to their colony tank, and all the eggs from the same strain from the breeding tanks were pooled and maintained at 28 °C in plastic Petri dishes. The water was replaced with fresh aquarium water on days 1, 3, 4, and 5, and dead eggs and larvae were removed [8].

### 2.2. Experimental Protocol

Each zebrafish larva (6–7 d post-fertilization) was placed in a 96-well assay plate (AGC Techno Glass Co., Ltd., Shizuoka, Japan) containing 300 µL of an aqueous solution of breeding water and acclimatized to a dark place for 60 min to a video tracking apparatus (Noldus, Wageningen, The Netherlands). The zebrafish larvae were then exposed to 50 alternating light–dark cycles, where the time period of each illumination phase was 15 min or 5 min, and their locomotor responses were monitored. The light intensity used during the experiments was 5000 lux under artificial lighting and 0.1× under infrared illumination.

Statistical analysis of these results led to the shortest and more stable state.

### 2.3. Statistics

Group data were expressed as mean ± standard error of the mean. The data were analyzed using the Dunnett’s test, Tukey–Kramer test, or Student’s *t*-test at a confidence level of 95%. One-way analyses of variance were used for statistical comparisons of the recorded observational data, followed by pairwise post hoc comparisons. Statistical analysis was performed in GraphPad Prism 6 for Windows version 6.05 (GraphPad Software, San Diego, CA, USA).

## 3. Results

### 3.1. Light–Dark Tests with 15-Min Intervals

Based on past reports, the larvae were left to acclimatize for 1 h, and the light–dark test was repeated 50 times at 15-min intervals (Figure 1a) [1]. To specify the number of statistical stabilizations, the 1st, 10th, 20th, 30th, 40th, and 50th repetitions were arbitrarily selected (Figure 1b). Stability was achieved after the 20th repetition (Figure 1b). From the results of the Tukey–Kramer tests (Figure 1c), at the time of light-on, all 10 to 50 repetitions were compared with the first stimulus. The responses to the 20th and 40th stimuli were compared to that elicited by the 10th stimulus, and the responses to the 20th, 30th, and 40th stimuli were compared to that elicited by the 50th stimulus. At the time of light-off, the responses to the 10th to 50th stimuli were compared to that of the first stimulus. For the 10th repetition, statistically significant differences were tested after the 20th repetition. Next, to obtain a statistically stable minimum value, the 10th to 20th repetitions were extracted and tested using Dunnett’s test and the 20th repetition as a reference. As a result, statistically significant differences were detected between the 10th and 12th repetitions at the time of light-on. At the time of light-off, statistically significant differences were detected between the 10th and 15th repetitions (Figure 1d). Therefore, the minimum number of repetitions required to achieve statistical stability was 16.

The distances traveled by zebrafish larvae from the 10th to 20th repetitions were analyzed every minute for a more detailed assessment of this variable (Figure 2a). From the 16th to the 18th repetitions, zebrafish larvae were almost immobile when the light was on; when the light was off, the distance traveled in the first minute was 71.8 ± 4.4 mm, but then the larvae gradually became immobile (Figure 2b).

### 3.2. Light–Dark Tests with 5-Min Intervals

When the light–dark test was conducted at intervals of 15 min and the traveled distance was evaluated every minute, the movement distance did not change when the light was on; however, when the light was off, the larvae moved in the first minute and gradually became immobile (Figure 1 and Figure 2). Therefore, the larvae were allowed to stand for 1 h for habituation, and the light–dark test was repeated 50 times at 5-min intervals (Figure 3a). To specify the number of statistical stabilizations, the 1st, 10th, 20th, 30th, 40th, and 50th repetitions were arbitrarily extracted, and stability was achieved after the 10th repetition (Figure 3b). From the Tukey–Kramer test results (Figure 3c), all 10th, 30th, 40th, and 50th intervals were compared to the first stimulus at the time of light-on and light-off. Next, to obtain a statistically stable minimum value, the first to 10th stimuli were extracted and tested using Dunnett’s test with the 10th repetition as a reference. No statistically significant differences were detected in the response of zebrafish larvae exposed to light. At the time of light-off, statistically significant differences were detected between the first and fourth times (Figure 3d). Therefore, the minimum number of repetitions required to achieve statistical stability was five.

The distances traveled by the zebrafish larvae from the first to the 10th repetitions were analyzed every minute for a more detailed assessment of this variable (Figure 4a). From the 5th to the 7th repetitions, the zebrafish larvae did not move when the light was on, but the traveled distances increased when the light was off (Figure 4b).

## 4. Discussion

### 4.1. Fear and Anxiety in Fish

Fear and anxiety are indispensable defensive responses for the survival of animals, and a growing number of studies on these phenomena have been performed on fish [9]. A fearful reaction typically occurs when an animal directly faces a predator or senses a stimulus that indirectly indicates the presence of a predator, while an anxious reaction is caused by the possibility of encountering a predator [10,11]. Typical tests for assessing anxiety levels include open field tests, light/dark box tests, and novel tank tests. Open-field tests are used to assess how fish react when exposed to new anxiety-inducing environments [12]. The light/dark box test evaluates the responses of fish in an aquarium with a black and white background, taking advantage of the fact that fish feel safer in dark environments [13]. The novel tank test determines the preference for the top or bottom half of the aquarium as an indicator of anxiety [14].

Tests aimed at studying responses to fear typically involve alarm stimuli; that is, a reaction to an aversive stimulus such as a sound, an optical signal, or an electric shock.

There are notable differences between the fear and anxiety responses [9,15]. Fear is a transient state that occurs when detecting or foreseeing the current or imminent threats (such as predators), and it disappears as soon as the object of fear disappears. Anxiety is caused by unspecified, uncertain, and unpredictable threats, and is a more persistent condition than fear. However, in fish research, the distinction between the two is not always clear. This phenomenon may influence the behavior of fish in light–dark tests and cause observational misinterpretation as light and dark stimuli become predictable when repeated regularly, as in some light–dark tests.

### 4.2. Light–Dark Stimulation Test of Zebrafish Larvae

Behavioral observations of unstimulated zebrafish may remain unchanged, as evidenced by the 1-h adaptation time of this experiment, and zebrafish juveniles are sensitive to various stimulus modalities, such as touch, smell, chemical sensation, hearing, vestibular and visual input, and heat [16]. Their response to stimuli can be observed within a short time. Stimulation methods include hammering stimulus [17] and light–dark stimuli [1,7,18,19], with the latter providing highly sensitive and stable data in neuropharmacology [1]. Though many studies have used light and dark stimuli [1,7,8,18,19], none have thoroughly examined the effects of the time interval and number of repetitions of stimuli on the behavior of zebrafish.

Understanding the response of zebrafish to their visual environment is important for building a behavioral experiment system. For example, the lighting and color of the aquarium have a great influence on the behavior of zebrafish. Therefore, Facciol et al. (2017) proposed that it is necessary to distinguish between the “difference in lighting intensity” and the “background color of the aquarium” to define light and darkness [20]. In this study, the “background color of the aquarium” was kept constant, and an evaluation based on the “difference in light intensity” was adopted.

Zebrafish offer several advantages as model animals for behavioral studies. Their larvae show a clear swimming pattern that depends on light/dark conditions after the swim bladder develops 4 d after fertilization [1].

In addition to the technical aspects, the use of zebrafish larvae follows the 3 Rs approach—replacement, reduction, refinement [21,22]—which strives to reduce the use of living animals in biomedical research [23].

### 4.3. Number of Light and Dark Stimuli Repetitions

Habituation is usually a non-associative form of learning, where the response to a repeated stimulus gradually weakens [24]. This phenomenon is evolutionarily conserved and present in a wide range of species, from invertebrates to vertebrates, such as rodents [24,25]; it acts as a mechanism by which the nervous system filters irrelevant stimuli. Zebrafish larvae at 6 d after fertilization have been demonstrated to show habituation [1]. Our study results demonstrated that when light and dark stimuli were repeated 50 times successively at 15-min intervals, the amplitude of the traveled distance gradually decreased and stabilized. This could demonstrate the process of a habituated response. In contrast, the light–dark stimulus at 5-min intervals did not attenuate the movement of zebrafish larvae, and the amplitude of the traveled distance did not decrease.

To understand these responses better, we scrutinized every minute of the repetitions when the traveled distance was estimated to stabilize (16th repetition at 15-min intervals, 5th repetition at 5-min intervals). With the light–dark stimulus at 15-min intervals, the traveled distance reached its peak and gradually declined in the first minute of the darkened stimulus. Then, in approximately 8 min, it almost returned to the stationary baseline. From these results, to investigate the moving time effectively, light and dark stimulation was performed at 5-min intervals after the zebrafish larvae started moving again. At the 5-min light–dark interval, the movement distance gradually started to rise, and at the 5th min, the movement distance reached the peak again. Then, the movement distance decreased due to light stimulation. This tendency did not change even when arbitrary repetitions (1st, 10th, 20th, 30th, 40th, and 50th) were extracted and compared (Appendix A).

These results do not clearly imply that zebrafish larvae constantly move in dark environments, and it must be noted that the timing of light stimulation also plays an important role. Additionally, the traveled distance stabilizes at the 16th repetition of the light/dark stimuli at 15-min intervals, and it takes 9 h to complete the 16 repetitions. In contrast, the traveled distance stabilizes at the 5th repetition of the light/dark stimuli at 5-min intervals, which only takes 1 h and 50 min to complete. In this study, the time of habituation to the study conditions was set to 1 h. Considering this acclimation time, it is expected that the time required to obtain more stable data will be shortened. Although more detailed data were not included in this study, we examined the effects of different time intervals of light/dark stimulation, specifically, intervals of 10 min, 3 min, and 1 min (Appendix A). The traveled distance of zebrafish took longer to stabilize in the 10-min intervals than in the 5-min intervals. Sufficient travel distance was not obtained for the 3-min intervals, and the data varied as a consequence. The 1-min intervals varied from start to finish. Although we did not analyze any other intervals of time, it seems that light/dark stimuli at 5-min intervals are appropriate.

## 5. Conclusions

Although many behavioral experiments were conducted on zebrafish using light and dark stimuli, there are no reports on specific protocols. When evaluating the habituation times, as indicated by the stabilization of the distance traveled by zebrafish, the distance was only stabilized at the 16th repetition of the stimulus when the zebrafish were subjected to light–dark stimuli at 15-min intervals. In contrast, when subjected to light and dark stimuli at 5-min intervals, the distance was initially stabilized at the 5th repetition. Although the acclimation time was set to 1 h in both cases, the fish took 9 h during the 15-min interval and 1 h and 50 min during the 5-min interval to become habituated to the stimuli. To shorten the habituation time, it is necessary to consider the time required for acclimation.

The zebrafish used this time is a wild type zebrafish that is widely sold for research purposes. I strongly hope that many researchers, including the authors, can use this protocol.

In conclusion, to obtain habituation data efficiently, it is advisable to evaluate the fifth repetition of light and dark stimulations at 5-min intervals when conducting experiments.

## Figures and Tables

**Figure 1 biomedicines-09-00884-f001:**
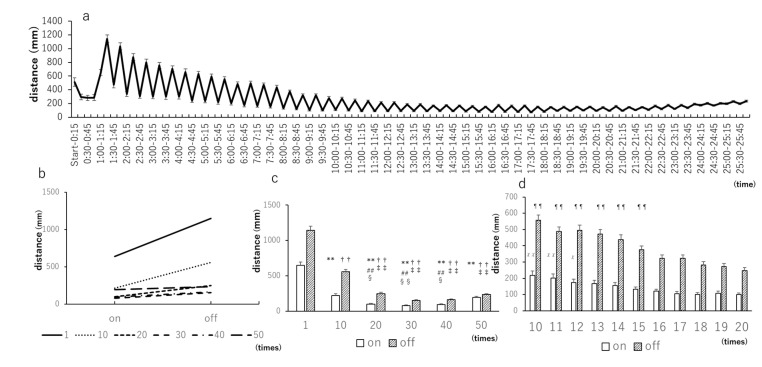
Light–dark test with 15-min intervals repeated 50 times. (**a**) Distance traveled by zebrafish larvae when light and dark stimuli were alternately repeated 50 times at 15-min intervals after being acclimated to a dark place for 1 h. As the number of repetitions increases, the traveled distance becomes smaller and more constant; (**b**) Distance traveled when the light is on and off at arbitrary numbers of repetitions (1st, 10th, 20th, 30th, 40th, and 50th). At all points, the traveled distance is larger when the light is off than when the light is on. After the 20th repetition, there are almost no changes in the traveled distance; (**c**) Tukey–Kramer test results; (**d**) Dunnett’s test results, where the 10th to 20th repetitions were extracted and tested with the 20th repetition as reference. ** *p* < 0.01 vs. 1 (on), ^##^
*p* < 0.01 vs. 10 (off), ^§^
*p* < 0.05 vs. 50 (on), ^§§^
*p* < 0.01 vs. 50 (on), ^††^
*p* < 0.01 vs. 1 (off), ^‡‡^
*p* < 0.01 vs. 10 (off), ^χ^
*p* < 0.05 vs. 20 (on), (Tukey–Kramer test). ^χχ^
*p* < 0.01 vs. 20 (on), ^¶¶^
*p* < 0.01 vs. 20 (off), (Dunnett’s test). Values are presented as mean ± SE (*n* = 47).

**Figure 2 biomedicines-09-00884-f002:**
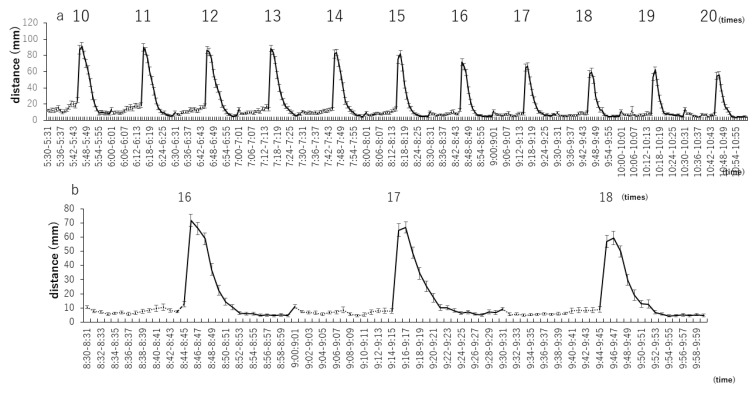
Distance traveled by zebrafish larvae in 1 min with specific numbers of repetitions of light–dark stimuli at 15-min intervals. (**a**) Distances traveled by zebrafish larvae from the10th to the 20th repetitions were analyzed every minute for a more detailed assessment of this variable. (**b**) Analysis of stable 16th to 18th distance traveled.

**Figure 3 biomedicines-09-00884-f003:**
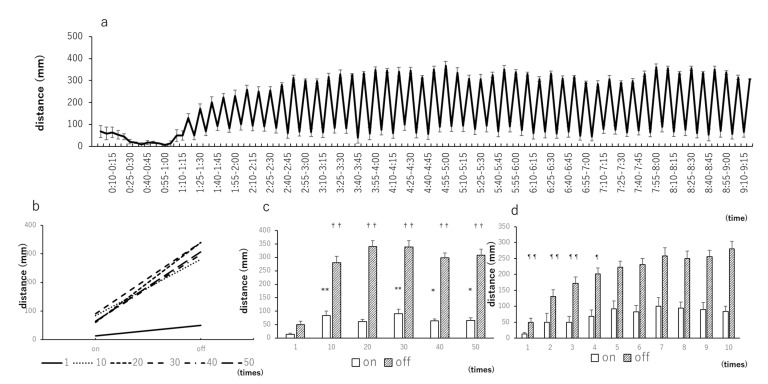
Light–dark test with 5-min intervals repeated 50 times. (**a**) Distance traveled by zebrafish larvae when light and dark stimuli were alternately repeated 50 times at 5-min intervals after being acclimated to a dark place for 1 h. As the number of repetitions increases, the traveled distance becomes smaller and more constant; (**b**) distance when the light is on and off at arbitrary numbers of repetitions (1st, 10th, 20th, 30th, 40th, and 50th). At all points, the traveled distance is larger when the light is off than when the light is on. After the 10th repetition, there are almost no changes in the traveled distance; (**c**) Tukey–Kramer test results; (**d**) Dunnett’s test results, where the 1st to 10th repetitions were extracted and tested using the 10th repetition as reference. * *p* < 0.05 vs. 1 (on), ** *p* < 0.01 vs. 1 (on), ^††^
*p* < 0.01 vs. 1 (off) (Tukey–Kramer test). ^¶^
*p* < 0.05 vs. 10 (off), ^¶¶^
*p* < 0.01 vs. 10 (off) (Dunnet’s test). Values are presented as mean ± SE (*n* = 42).

**Figure 4 biomedicines-09-00884-f004:**
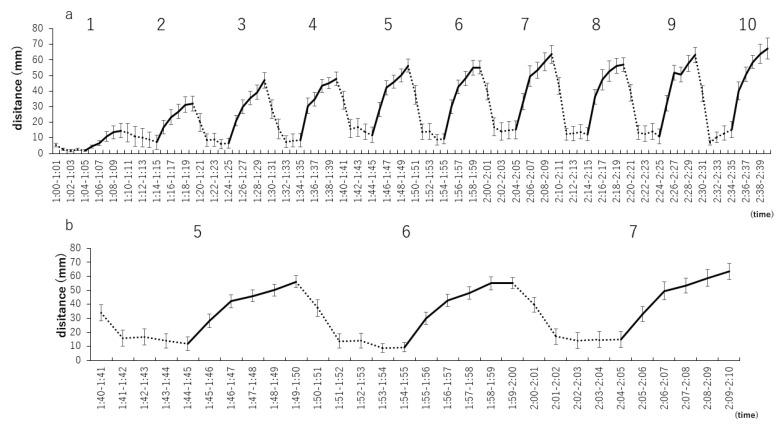
Distance traveled by zebrafish larvae in 1 min with specific numbers of repetitions of light–dark stimuli at 5-min intervals. (**a**) Distances traveled by zebrafish larvae from the first to the 10th repetitions were analyzed every minute for a more detailed assessment of this variable. (**b**) Analysis of stable 5th to 7th distance traveled.

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
