# Peer review of "Standardized Method for the Assessment of Behavioral Responses of Zebrafish Larvae"

_biomedicines, 2021, doi:10.3390/biomedicines9080884_

Round 1

Reviewer 1 Report

Maeda et al. describe in their manuscript entitled „Standardized method for the assessment of behavioral responses of zebrafish larvae“ responses of 6-7 dpf larvae in terms of light-dark stimulus induced movements. The paper is well written and the conducted experiments and analyses are in general well described. Hasumi and co-workers studied the responses of zebrafish larvae following different intervals of light-dark stimuli with the goal to define an optimal setup to study behavioral responses in zebrafish larvae. The overall conclusion – with 5-min intervals being most suitable to obtain reliable results – is understandable based on the presented data sets. I also appreciate that the authors were aiming at a standardized protocol for such experiments; although zebrafish are increasingly recognized as an animal model in pharmaceutical research there are only few reports suggesting standardized methods, which would be required to further increase popularity and acceptance of this model.

Despite the relevance of the presented study, I have some minor and some major revision points:

  • As this study aims at a standardized protocol, it is required the also assay conditions are standardized. In this respect, the authors should give more details on embryo/larvae handling. Which line was used? Why did the authors use system water for keeping larvae instead of a standardized and commonly used buffer (e.g. 0.3x Danieau’s solution)? This may have a significant impact on the reproducibility of the presented results when experiments are done in another laboratory.
  • What type of 96-well MTPs was used? 300 µL/well appears to exceed the typical working volume of standard MTPs.
  • The number of biological/technical replicates should be given in a more obvious way in the methods section and/or figure captions.
  • Line 186ff: references are not in brackets [..]
  • The presented study would benefit a lot from adding the assessment of CNS active reference drugs. By this, the authors could support their findings and conclusion of 5-min intervals being most suitable compared to longer and shorter intervals. In particular, it remains unclear whether the data quality of drug screening could be indeed improved when implementing their proposed protocol.

Reviewer 2 Report

Authors provided a paper entitled “standardized method for the assessment of behavioral responses of zebrafish larvae” for the publication in Biomedicines, a Journal of MDPI Publisher.

The paper is characterized by a good scientific interest, but it requires some English revisions so far.

For example, I suggest these modifications in the manuscript text:

Line 11. “since” or “because of” in the place of just “because” to start this sentence.

Line 12. Addition of “using” among “observed” and “Multiple rapid…”

Line 16. “has been increased” in the place of “increasing”.

Line 20. “6-7 days” in the place of “d”.

line 31. Clarify “lethal concentrations of unknown substances”.

Line 35. “Dario rerio should be in italique

Line 37. “in vitro” should be in italique.

Line 60. “Basnet et al. (2019)”. maybe better to uniform this reference, using square parenthesis.

Introduction is clear, but the state of the art could be enlarged, also citing more references.

The scope of the paper is slightly cited among lines 66 and 68; in my opinion the aims should be better presented and argumentations should be also provided.

Clear description of the animals; however, descriptions of the protocols could be enlarged.

Results

I do not see Figure 1a recalled in the manuscript, only figure 1b, 1c and 1d.

Line 124. could you define “almost immobile”; this means that the distance was less than ?

Line 125, what does it mean that the “distance was large”. please quantify.

Line 98 and Line 137. Could be useful to link a mathematical model to Tukey and Kramer test?

Concerning data of Figure 1a, would it be possible that the movements described in the dark hours, are just randomic Brownian movements?

Line 183. “a growing number of studies on this topic have been performed on fish”, but no studies are cited after this sentence.

Line 187. Open field could be described in the material and methods section.

Line 205. “considerable amount of time”, please define how much

What is affirmed in paragraph 220-222, does it find any confirmation in the literature? if yes, cite them.

Future perspectives could be added to the conclusions section

thank you.

Reviewer 3 Report

I have suggestion for optimisation of structure of the MS and several minor remarks:

  • line 35. The Latin name of the fish under study is Danio (not Dario). Additionally, the Latin name of living organisms is commonly provided in Italics in biological literature;
  • line 38. Be careful with statement that “morphology of fish is highly homologous of those of mammalian species”;
  • line 52. “Light-dark” instead of “Light-dart”;
  • lines 66-68. The aim of the investigation is better as: “The aim of this study was to find an optimal experimental method convenient for analysis of responses of zebrafish larvae to light and dark stimuli”;
  • line 71. … organizations?
  • line 73. “aquarium water” is not quite understandable. Was it tap water?
  • line 80. “eggs” is more appropriate here instead of “embryos”
  • line 180 and below. The Discussion section contains text (e.g., 4.1.) more appropriate for Introduction because it discusses general questions without direct referring of the obtained results. On the other hand, text of the second paragraph of 4.3. looks like results. The structure of the manuscript should be slightly reorganized;
  • line 276. “acclimation” instead of “acclimatization” is better in this case.
  • lines 277-279. The last sentence of Conclusion is not appropriate. It confirm that more optimal experimental protocols are possible; thus, the authors report uncompleted research;
  • lines 289-290 and 295-297. The text of “Author Contribution” section includes text from Guidelines for Authors.
  • lines 305-312. This text from Guidelines for Authors.
  • the same is true for lines 313-317 (“Data Availability Statement” section).

Round 2

Reviewer 1 Report

The authors addressed most of my revision points. However, I  still have several concerns, which are mostly related to the fact that the provided manuscript should serve as a basis for standardizing zebrafish screening:

  • The used zebrafish line is not specified in detail (I believe most labs use the AB line, so the authors should consider either identifying their line or adding an additional experimental data set with a well-defined and broadly used line). Please also add more details on husbandry of the adult fish.
  • I still disagree with keeping the embryos in system water prior to the behavioral studies. I really appreciate the fact that the authors were aiming at a standardized procedure for such studies, however, reproducibility between labs seems problematic as other labs might have completely different water conditions in their system. In that respect, it would be beneficial to collect embryos in a defined embryo medium prior to subjecting them to the assay at 6-7 dpf.
  • Along the same lines, I still think that adding some reference drugs would improve the manuscript. If the authors aim at having their protocol applied routinly by other labs, I would find it more convincing if they could show that the 5-min intervals are indeed giving better/more consistent data. However, this might be part of a follow-up study. 

Reviewer 2 Report

Authors proposed the second version of their paper

Authors responded to my issues point by point. The paper is improved and deserves to be published

A good number of interventions have been performed in the manuscript.

Information about manufacturer and countries were added properly.

I suggest to eliminate the written test “Figure 1” embedded in the figure, since there is already Figure 1. there are two lines captions embedded in the figure cluster, that should need to be enlarged or incorporated in the real Figure 1 caption.

Thank you.

Round 3

Reviewer 1 Report

I am mostly fine with the revised version and the repsonse by the authors.

The following points should be addressed during revision:

  • Re: the used zebrafish line; the additionally provided source data are fine. But I would recommend to the authors to use a better characterized line (e.g. AB) in future studies. This might be also mentioned/briefly discussed in the conclusion section
  •  Re: system water for embryo maintenance; I do not fully understand the reply by the authors ("We have obtained the same results in duplicate experiments. In other words, the results have been the same and the time has been significantly reduced") My concern was related to reproducibility between different labs (here's again my initial comment: "I still disagree with keeping the embryos in system water prior to the behavioral studies. I really appreciate the fact that the authors were aiming at a standardized procedure for such studies, however, reproducibility between labs seems problematic as other labs might have completely different water conditions in their system. In that respect, it would be beneficial to collect embryos in a defined embryo medium prior to subjecting them to the assay at 6-7 dpf)." This should be addressed or critically discussed.
  • Re: reference drugs for validation; I suggested to use reference drugs (CNS actives) to strengthen the study. The authors replied that "The same results have been obtained in our existing experiments. In other words, the results are the same and the time is significantly reduced." If such data exist these should be added to the manuscript.

Author Response

Please see the attach file.
